# Land Suitability Assessment and Agricultural Production Sustainability Using Machine Learning Models

**Ruhollah Taghizadeh-Mehrjardi** [1,2], **Kamal Nabiollahi** [3,*]**, Leila Rasoli** [3]**, Ruth Kerry** [4] **and Thomas Scholten** [1,5,6]

[1] Department of Geosciences, Soil Science and Geomorphology, University of Tübingen, 72070 Tübingen, Germany; ruhollah.taghizadeh-mehrjardi@mnf.uni-tuebingen.de (R.T.-M.); thomas.scholten@uni-tuebingen.de (T.S.)
[2] Faculty of Agriculture and Natural Resources, Ardakan University, Ardakan 8951656767, Iran
[3] Department of Soil Science and Engineering, Faculty of Agriculture, University of Kurdistan, Sanandaj 6617715175, Iran; leila.16665rasouly@gmail.com
[4] Department of Geography, Brigham Young University, Provo, UT 84602, USA; ruth_kerry@byu.edu
[5] CRC 1070 ResourceCultures, University of Tübingen, 72070 Tübingen, Germany
[6] DFG Cluster of Excellence "Machine Learning", University of Tübingen, 72070 Tübingen, Germany
[*] Correspondence: k.nabiollahi@uok.ac.ir

**Abstract:** Land suitability assessment is essential for increasing production and planning a sustainable agricultural system, but such information is commonly scarce in the semi-arid regions of Iran. Therefore, our aim is to assess land suitability for two main crops (i.e., rain-fed wheat and barley) based on the Food and Agriculture Organization (FAO) "land suitability assessment framework" for 65 km$^2$ of agricultural land in Kurdistan province, Iran. Soil samples were collected from genetic layers of 100 soil profiles and the physical-chemical properties of the soil samples were analyzed. Topography and climate data were also recorded. After calculating the land suitability classes for the two crops, they were mapped using machine learning (ML) and traditional approaches. The maps predicted by the two approaches revealed notable differences. For example, in the case of rain-fed wheat, results showed the higher accuracy of ML-based land suitability maps compared to the maps obtained by traditional approach. Furthermore, the findings indicated that the areas with classes of N2 (≈18%↑) and S3 (≈28%↑) were higher and area with the class N1 (≈24%↓) was less predicted in the traditional approach compared to the ML-based approach. The major limitations of the study area were rainfall at the flowering stage, severe slopes, shallow soil depth, high pH, and large gravel content. Therefore, to increase production and create a sustainable agricultural system, land improvement operations are suggested.

**Keywords:** random forests; support vector machine; parametric method; rain-fed wheat; barley

## 1. Introduction

Rapid population growth in developing countries means that more food will be required to meet the demands of growing populations. Rain-fed wheat and barley, as major grain crops worldwide, are planted under a wide range of environments and are a major staple source of food for humans and livestock [1–4]. The production of such staple crops influences local food security [5]. Rain-fed wheat and barley are cultivated on approximately 6 and 0.64 million ha in Iran, respectively [6]. They are well adapted to the rain conditions of western Iran, where mean precipitation is 350–500 mm. The production of rain-fed wheat and barley per unit area in Iran is low compared to developed

countries worldwide [2]. One of the main causes for this low yield is that the suitability of land for their cultivation has not been determined. To overcome this problem, land suitability assessment is needed, which can help to increase crop yield by growing these crops in the locations that are most suited to their growth [7].

The first step in agricultural land use planning is land suitability assessment which is often conducted to determine which type of land use is suitable for a particular location [8]. Land suitability assessment is a method of land evaluation, which identifies the major limiting factors for planting a particular crop [9,10]. Land suitability assessment includes qualitative and quantitative evaluation. In the qualitative land suitability evaluations, information about climate, hydrology, topography, vegetation, and soil properties is considered [11] and in quantitative assessment, the results are more detailed and yield is estimated [12]. The FAO land evaluation framework [13,14] and physical land evaluation methods [15] have been widely used for land suitability assessment.

Land suitability maps provide the necessary information for agricultural planners and are vital for decreasing land degradation and for assessing sustainable land use. There is a lack of land suitability mapping and associated information in Iran because land suitability surveying and mapping in Iran have followed the traditional approach [16–20]. In the traditional approach, abbreviation of the soil variability through a soil map unit to a representative soil profile may cause the precision of the land suitability maps to be lacking and ignores the continuous nature of soil and landscape variation, resulting in the misclassification of sites and discrete and sharply defined boundaries [21,22]. Moreover, the traditional approach is time-consuming and costly [23].

Machine learning (ML) models are capable of learning from large datasets and integrate different types of data easily [24,25]. In digital soil mapping framework, these ML models have been applied to make links between soil observations and auxiliary variables to understand spatial and temporal variation in soil classes and other soil properties [24,26–28]. These ML models include artificial neural networks, partial least squares regressions, support vector machines, generalized additive models, genetic programming, regression tree models, k nearest neighbor regression, adaptive neuro-fuzzy inference system, and random forests [26–28]. It should be noted that random forests and support vector machines have been the most commonly used techniques in the digital soil mapping community in the last decade due to their relatively good accuracy, robustness, and ease of use. The auxiliary variables can be obtained from digital elevation models (DEM), remotely sensed data (RS), and other geo-spatial data sources [24,29–35].

Although in recent years, ML models have been widely used to create digital soil maps [24], little attempt has been made for using ML models to digitally map land suitability classes [36,37]. For instance, Dang et al. [38] applied a hybrid neural-fuzzy model to map land suitability classes and predict rice yields in the Sapa district in northern Vietnam. Auxiliary variables included eight environmental variables (including elevation, slope, soil erosion, sediment retention, length of flow, ratio of evapotranspiration to precipitation, water yield, and wetness index), three socioeconomic variables, and land cover. Harms et al. [39] assessed land suitability for irrigated crops for 155,000 km$^2$ of northern Australia using digital mapping approaches and machine learning models. They concluded that the coupling of digitally derived soil and land attributes with a conventional land suitability framework facilitates the rapid evaluation of regional-scale agricultural potential in a remote area.

Although Kurdistan province is one of the main agriculturally productive regions of Iran and holds an important role in the country's crop production rank, the mean yield of rain-fed wheat and barley in these regions is lower than 800 kg ha$^{-1}$ [40]. Land suitability maps can classify the areas that are highly suitable for the cultivation of the two main crops and can help to increase their production. However, such information is commonly scarce in these semi-arid regions. Therefore, the main objective of this study is to assess the land suitability for two main crops based on the FAO "land suitability assessment framework". Furthermore, in this study, we focus on using machine learning models to predict the spatial distribution of land suitability classes in the most economically feasible way and explore if it works better than the traditional approach—the most common approach to

produce land suitability class maps in Iran. Two machine learning models algorithms, random forest (RF) and support vector machine (SVM), were selected due to their successful applications in earlier studies [29–35] and their relatively good accuracy, robustness, and ease of use. Importantly, it has been shown that both RF and SVM models work well when soil data are not particularly available. Additionally, it is important in this work to explain the complex relationships between the predicted land suitability class and the auxiliary variables, which can be inferred by exploring the importance of auxiliary variables.

## 2. Materials and Methods

### 2.1. Site Description

The study area is located in Kurdistan province, western Iran. It surrounds the city of Ghorveh and covers a region of 65 km$^2$ (Figure 1). The climate is semi-arid with a cold and rainy winter and a moderate and dry summer. The mean yearly rainfall is 369.8 mm and over 90% of the rain falls between November and March. The mean annual temperature (10.8 °C) is relatively cool. Soil moisture and temperature regimes are Xeric and Mesic, respectively. Elevation varies between 1833 and 2627 m above sea level and the area is surrounded by mountains and hills from the southwest to the southeast. The two main land uses of the study area are cropland (approximately 60%) and rangeland, where rain-fed winter wheat and barley are the most important crops in the study area. (Figure 1). The geomorphologic units include piedmont, fan, hill, and mountain (based on the method presented in Toomanian et al. [41] and slope varies from gentle to very steep.

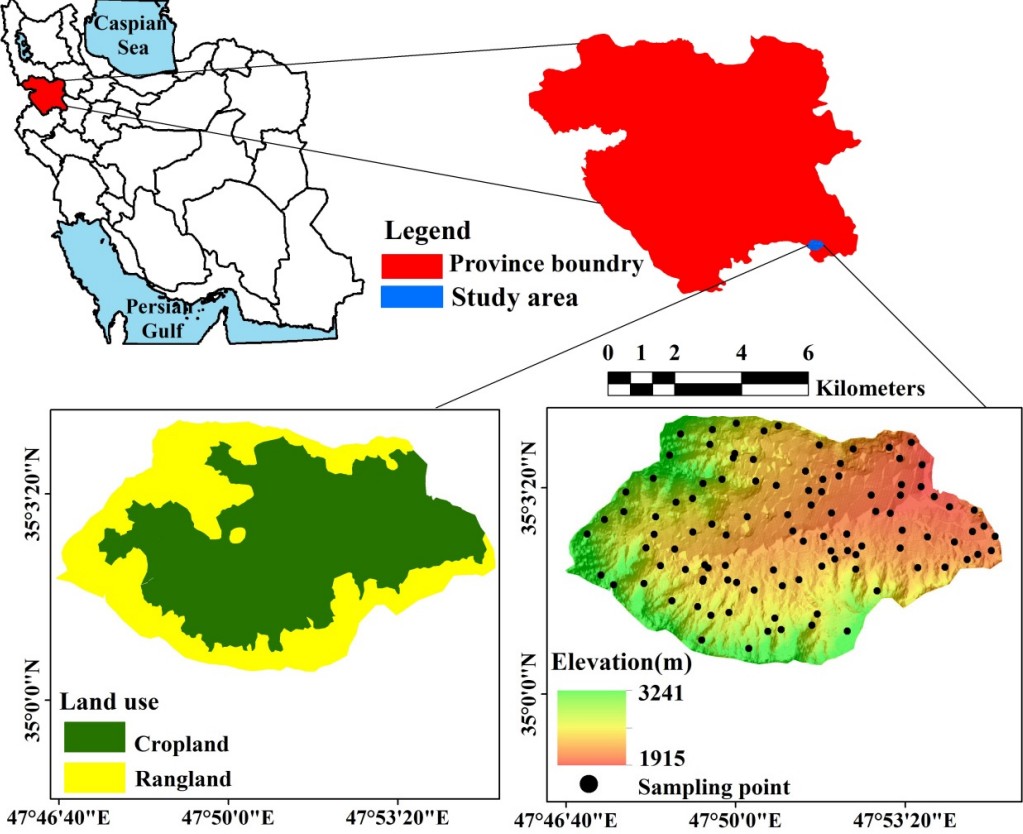

**Figure 1.** Location map of the study area within Kurdistan province and Iran.

### 2.2. Procedures

There were several stages to the analysis for this work and a flowchart of the procedures used in this research is shown in Figure 2. This work was conducted in several stages:

Selecting the locations of 100 soil profiles and collecting soil samples;

Analyzing the physical-chemical properties of the soil samples;

Collecting the topography and climate parameters;

Determining numeric ratings of soil, topography, and climate parameters [15,42,43];

Calculating land suitability index of rain-fed wheat and barley [44];

Calculating the land suitability class for rain-fed wheat and barley;

Preparing auxiliary variables at a regular grid spacing;

Determining a relationship between auxiliary variables and land suitability class using ML models;

Preparing a ML-based land suitability class map;

Preparing a traditional land suitability class map;

Comparing the ML-based and traditional land suitability class maps.

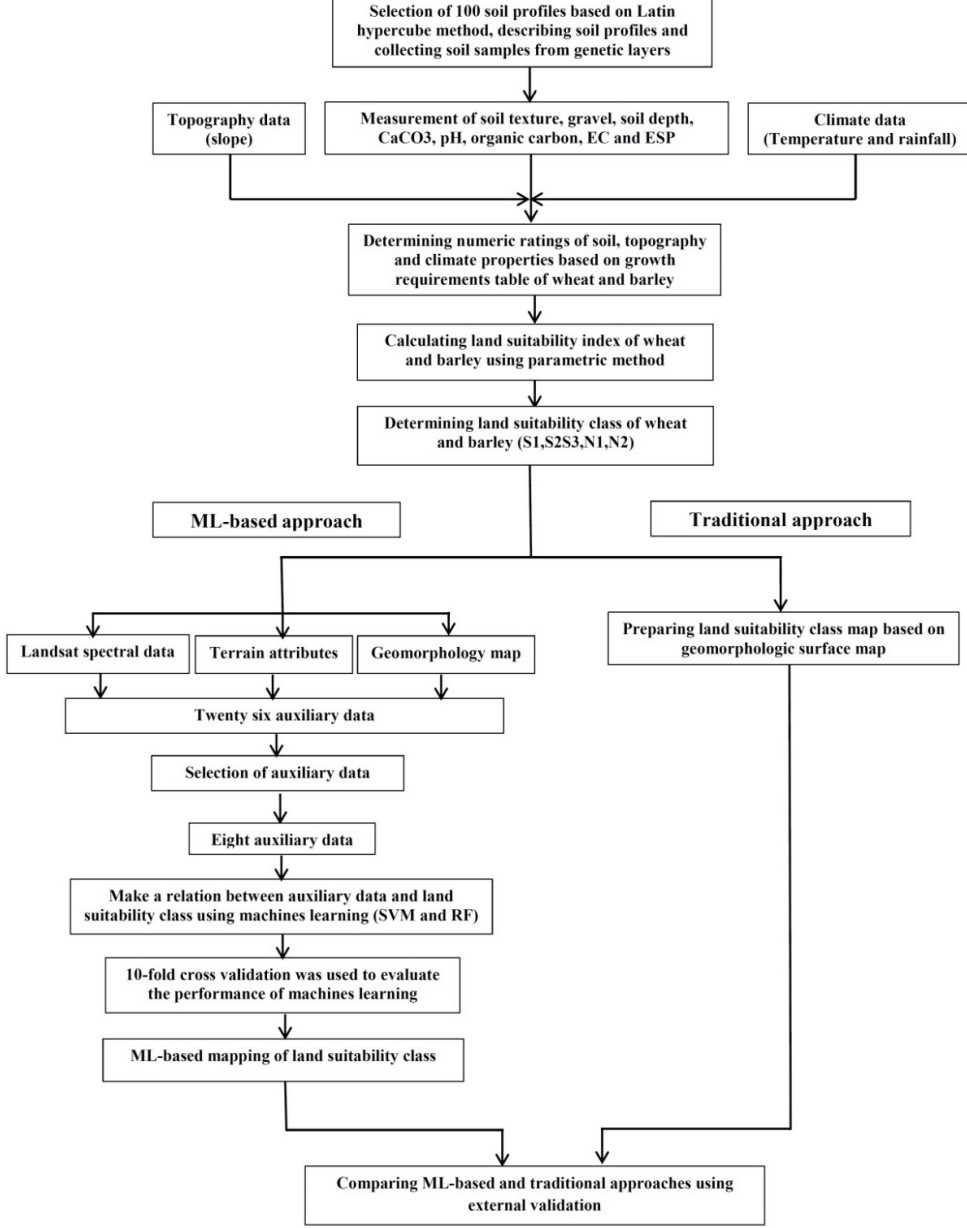

**Figure 2.** Flowchart of methodology used for machine learning (ML)-based and traditional land suitability assessment in this study. SVM: support vector machine; RF: random forest.

### 2.3. Data Collection and Soil Sample Analysis

In this study the locations of 100 soil profiles were assigned using the conditioned Latin hypercube [45] sampling method. Soil profiles were described [46] and soil samples were collected from genetic layers, air-dried at room temperature, and then passed through a 2 mm sieve prior to analysis of physical and chemical properties. Organic carbon was measured by wet combustion [47]. Soil pH and electrical conductivity (EC) were determined in a saturated paste by a pH electrode [48] and conductivity meter [49]. Cation exchange capacity (CEC) was determined by the 1 N ammonium acetate (at pH 7.0) method [50]. The calcium carbonate equivalent (CCE) was measured using a volumetric method [51]. The particle size distribution was determined by the Bouyoucos hydrometer method [52]. Exchangeable sodium percentage (ESP) was determined as the ratio of sodium, to CEC. Gypsum content was measured but as it was zero it was not considered further. As required, topography and climatic data for land suitability assessment were also obtained from a digital elevation model (DEM) [53] and from the Ghorveh synoptic meteorological station for the 30-year period 1987–2017, respectively.

### 2.4. Land Suitability Assessment

#### 2.4.1. Calculating Land Suitability Index Using Parametric Methods

In this study a qualitative assessment of land was performed to determine land suitability classes for rain-fed wheat and barley. The selection of influencing factors was done based on the growth requirement tables for rain-fed wheat and barley [42,43]. Climate (rainfall and temperature), slope, soil texture, gravel, soil depth, $CaCO_3$, soil pH, organic carbon, EC, and ESP were selected for calculating land suitability index. The selected parameters were scored based on [43]. The soil, topography, and wetness properties (e.g., gypsum, micro-relief, drainage, flooding, etc.) had no limitation for rain-fed wheat and barley (data not shown) so they were not considered in determining land suitability class. Base saturation percentage and cation exchange capacity (CEC) also were not considered for land suitability because the climate of the region is semiarid [15]. The average of the selected soil properties was determined by considering a depth-weighted coefficient up to a depth of 100 cm [15].

After the selection of parameters, the parametric method (square root method) was applied to determine the land suitability index of rain-fed wheat and barley. The methodology initially needs to evaluate climate. Therefore, the climatic index is calculated using a rating of climate characteristics. This climate index was converted to a climate rating and then along with the soil and topography ratings they were applied to calculate the final land suitability index. The square root method applies the following equation to calculate land suitability index (Equation (1)):

$$I = Rmin \sqrt{\frac{A}{100} \times \frac{B}{100} \times \frac{C}{100} \times \dots}$$

(1)

where *I* is the square root index, *Rmin* is the minimum rating, and *A*, *B*, *C*, are the other rating values [44]. In the parametric method, a numerical rating (0 to 100) is given to each soil property based on the level of limitation presented by each soil property (it is done in consultation with standard charts).

#### 2.4.2. Determining Land Suitability Class

After calculating land index for rain-fed wheat and barley, land suitability classes were determined. Land suitability classes include five classes [13]: highly suitable (S1), moderately suitable (S2), marginally suitable (S3), and non-suitable (N1 and N2). The land index of the N2, N1, S3, S2, and S1 classes ranged between 0–12.50, 12.50–25, 25–50, 50–75, and 75–100, respectively. Non-suitable (N) land was assumed to have severe limitations which could never be improved, or only marginally so through improvement practices.

### 2.5. Maps of Land Suitability Class

To prepare a map of land suitability classes across the study area, we implemented two approaches. In the first approach, the maps were prepared using ML models and auxiliary variables. In the second approach, the maps were prepared according the traditional approach, which is the most common approach to produce land suitability class maps in Iran.

#### 2.5.1. ML-Based Land Suitability Map

Two ML algorithms (i.e., SVM and RF) and a set of auxiliary variables (i.e., geomorphology, terrain attributes, and remotely sensed data) were used to predict land suitability classes of rain-fed wheat and barley. We selected two machine learning models including random forest (RF) and support vector machine (SVM) due to their successful applications in earlier studies [29–35] and their relatively good accuracy, robustness, and ease of use. Importantly, it has been proved that both RF and SVM work well when there is no massive availability of data. Additionally, it is important in this work to explain the complex relationships between the predicted land suitability class and the auxiliary variables, which can be inferred by the exploring the importance of auxiliary variables.

#### Grid-Based Auxiliary Variables

Geomorphology maps are useful auxiliary data as they contain useful information such as soil parent material and soil genesis [29,33,54,55]. In this study, geomorphological units were delineated into four levels including landscape, landform, lithology, and geomorphological surface based on aerial photos [41]. The ortho air photos were geo-referenced and the boundaries of the geomorphological surfaces that were delineated were inserted into a GIS environment (Figure 3).

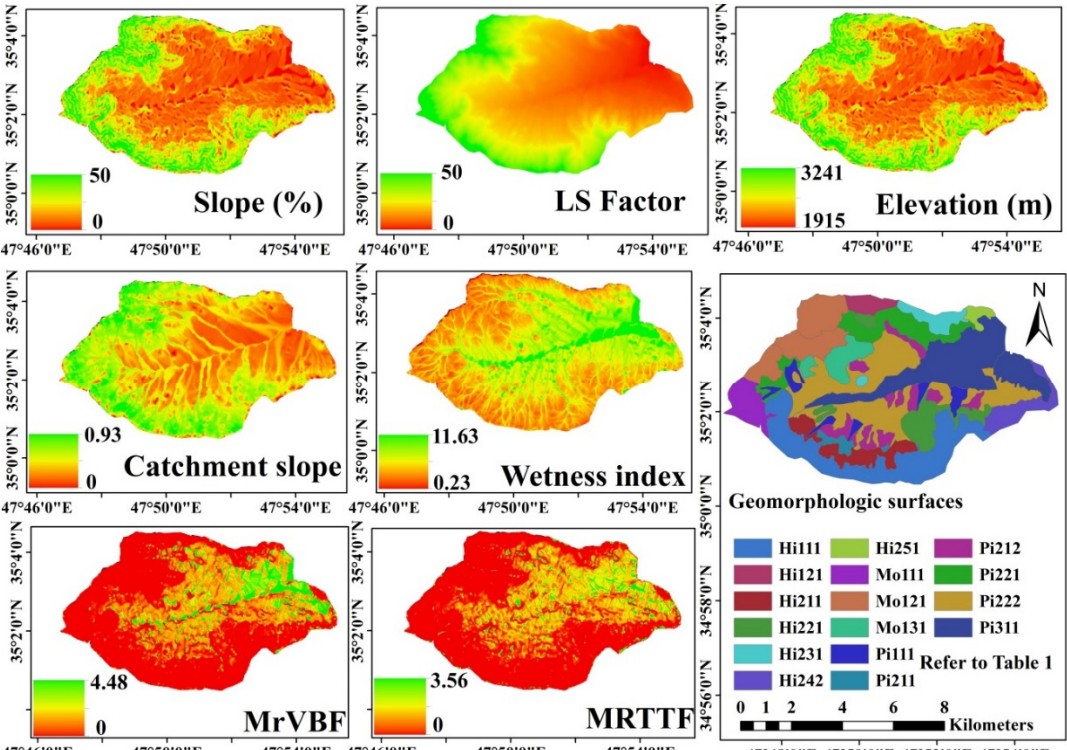

**Figure 3.** Eight most important auxiliary variables derived from Landsat spectral data and digital elevation models in the study area: MrVBF: multi-resolution valley bottom flatness index; MrRTF: multi-resolution ridge top flatness. (Geomorphic surfaces codes refer to Table A1 in Appendix A).

Terrain attributes including multi-resolution ridge top flatness (MrRTF), catchment network base level (CNBL), elevation (EL), catchment slope (CS), slope (SL), aspect (AS), length-slope factor (LS factor), plan curvature (PC), long curvature (LC), cross curvature (CC), valley depth (VD), topographic wetness index (TWI), and the multi-resolution index of valley bottom flatness (MrVBF) were extracted and computed through a digital elevation model (DEM) with a $10 \times 10$ m grid cell resolution [53] using SAGA GIS software (System for Automated Geoscientific Analysis) [56].

The following spectral bands and spectral indices were derived from a Landsat 8 OLI (operational land imager), acquired on 25 July 2017: spectral bands ((B1 (0.43–0.45 μm), B2 (0.45–0.52 μm), B3 (0.52–0.63 μm), B4 (0.63–0.68 μm), B5 (0.84–0.88 μm), B6 (1.56–1.66 μm), B7 (2.10–2.30 μm)), brightness index (BI, [57]), normalized difference vegetation index (NDVI, [58]), soil adjusted vegetation index (SAVI, [59]), and enhanced vegetation index (EVI, [60]). The spectral bands and indices were used as auxiliary variables for estimating land suitability class. All auxiliary variables are co-registered to the same raster grid with a size of $30 \times 30$ m.

Selection of Auxiliary Variables

In our study, we implemented the Boruta algorithm [61] with the RF classifier in the R statistical package [62] to rank the most relevant auxiliary variables for mapping land suitability classes. The Boruta algorithm is named after a god of the forest in Slavic mythology [61]. In short, this approach compares the importance of the auxiliary variables to five shadow variables. The values of those shadow variables are obtained by shuffling values of the auxiliary variables to remove their relationship with land suitability classes. The RF model uses the extended variables (i.e., auxiliary and shadow variables) to predict land suitability classes. Then, the Z-score, which is an indicator of the importance of all relevant auxiliary variables, is computed for each auxiliary variable and its corresponding shadow variable. Technically, the Z-score refers to the mean of accuracy loss divided by standard deviation of accuracy loss. Those auxiliary variables that scored better than the maximum Z-score are flagged as "important".

Machine Learning Models

Support vector machine (SVM) is a kernel method for classification [63] and regression problems [64]. The input data is transformed into a high dimensional feature space with a predefined kernel function. In high dimensional feature space, a linear regression hyperplane is derived for nonlinear relationships. Then, the hyperplane is back-transformed to nonlinear space. The kernel used in this study is a radial basis function. The e1071 package [65] was used for support vector machine modeling.

Random forest (RF) [66] is an ensemble technique based on classification and regression trees (CART) [67]. In respect to the response, CART uses binary splits to create more homogenous groups. For each tree of the ensemble, a bootstrap sample of the training data (instances) is used. Further, only random subsets of the auxiliary variables are used for the split at each node. The final tree outputs are averaged for the final prediction. A RF with a large number of trees is robust against overfitting, noise, non-informative, and correlated features [66]. The random forest package of [62] was used for RF training in this study.

Models Evaluation

Ten-fold cross validation with 100 replications was used to assess the ML models. In this study, two common performance metrics, namely kappa index and overall accuracy, were used. The performance metrics are functions of the confusion matrix as shown in Table 1.

**Table 1.** An example of a confusion matrix.

|  | **Actual Positive Class** | **Actual Negative Class** |
|---|---|---|
| Predicted positive class | True positive | False positive |
| Predicted negative class | False negative | True negative |

The overall accuracy is the ratio of all correctly classified land suitability classes to all used data. A higher overall accuracy indicates a high model performance (Equation (2)):

$$Overall\ accuracy = \frac{True\ Positive + True\ Negative}{True\ Positive + True\ Negative + False\ Positive + False\ Negative} \tag{2}$$

The kappa index is a robust index which takes into account the probability that a class is classified by chance [29]. It is a simple derived statistic that measures the proportion of all possible cases of presence or absence that are predicted correctly by a model after accounting for chance predictions. Similar to the overall accuracy, a higher kappa index indicates a high model performance [68,69] (Equation (3)):

$$Kappa\ index = \frac{1 - Overall\ accuracy}{1 - hypothetical\ probability\ of\ chance\ agreement} \tag{3}$$

### 2.5.2. Traditional Land Suitability Map

To develop the traditional land suitability maps, the geomorphology map of the study area was used as a base map (Figure 3). Finally, to compare the traditional and ML-based land suitability, we used an external validation technique based on 22 soil profiles (in addition to the 100 soil profiles used for prediction). The validation criteria used to determine the reliability of classes were the kappa index and overall accuracy.

### 2.6. Potential Yield

Potential yield is a crop production in an environment with no limitation of water, nutrients, and effective control of pests, diseases, and weeds. In other words, potential yield is defined plant growth under optimum conditions. Estimation of potential production has great importance in agricultural production management. In fact, potential yield represents the possibilities of production capital in different regions because of the difference between the potential yield and the actual average production of farmers in a region determines the amount of productivity in the use of inputs as the production capital. The lower difference (between the potential yield and the actual production) reflects the higher efficiency of inputs and closer to the possible production ceiling. Conversely, the higher difference reflects the greater need for research and development of agriculture.

In this research, potential yield of rain-fed wheat was computed by the FAO model [15] based on genetic potential plant and climate data such as radiation and temperature. The process contains following stages:

The first stage is calculation of bgm based on Equation (4):

$$bgm\ (kgCH2Oha-1\ hr-1) = f \times bo + (1-f)bc \tag{4}$$

where bgm is the maximum gross production of biomass ($kgCH_2Oha^{-1}\ hr^{-1}$), bo is the maximum gross production of biomass on overcast days ($kgCH_2Oha^{-1}\ hr^{-1}$), bc is the maximum gross production of biomass on clear days ($kgCH_2Oha^{-1}\ hr^{-1}$), and f is fraction of day time that the sky is overcast. The value of f can be as following:

The second stage is calculation of bn based on Equation (5):

$$Bn = \frac{(0.36 \times bgm \times KLAI)}{\left(\left(\frac{1}{L}\right) + 0.25 \times Ct\right)} \tag{5}$$

where Bn is the net production rate of biomass (kg ha$^{-1}$), ct is the respiratory rate that is gained from Equation (6), bgm is the maximum gross production of biomass (kgCH$_2$Oha$^{-1}$ hr$^{-1}$), KLAI is the correction factor for LAI (leaf area index) <5 m$^2$ m$^{-2}$, and L is the number of days required for production to be gained.

$$Ct = C30\left(0.044 + 0.0019 \times t + 0.001 \times t^2\right) \tag{6}$$

where Ct is respiration coefficient, C30 is the respiratory rate that for non-legume is 0.0108, t is the average temperature (C0).

The third stage is calculation of potential yield based on Equation (7):

$$Y = Bn \times Hi \tag{7}$$

where Y is potential yield (kgCH$_2$Oha$^{-1}$) and Hi is harvest index (ratio of grain yield to aboveground biomass).

## 3. Results and Discussion

### 3.1. Summary Statistics

Table 2 shows the descriptive statistics of soil properties in the study area. The average electrical conductivity is 0.44 dSm$^{-1}$ and its range is between 0.02 and 1.5 dSm$^{-1}$, which indicates the electrical conductivity is low. The average pH is 8.2 and its range is between 7 and 8.98, which show the soil pH is basic in the study area. The range of soil organic carbon changes from 0.66% to 2.6% and its average is 0.75%, which indicates soil organic carbon is low. The SOC exhibited a decreasing trend with depth. The calcium carbonate, ESP, and CEC data across the study area ranged from 0% to 34%, 0.34% to 12%, and 4.97 to 37.82 cmol$^+$kg$^{-1}$, respectively; and their averages are 17.09%, 2.97%, and 14.33 cmol$^+$kg$^{-1}$, respectively. The dominant soil texture classes in the region are loamy, clay loam, and sandy clay loam. It is worth noting that the gypsum content was zero in the study area. Based on the general evaluation to the coefficient of variation or CV value, the variation coefficients of SOC, EC, CCE, and gravel were high (more than 35%), which shows a high variability across the study area [70]. This large variation in SOC, EC, and CCE was due largely to the variability in topography and parent material. The coefficient of variation for other soil properties was moderate (between 15% and 35%). Therefore, soil properties have a broad range of values in the studied soils.

**Table 2.** Descriptive statistics for soil properties.

| | Unit | C.V | S.D | Maximum | Minimum | Mean |
|---|---|---|---|---|---|---|
| pH | | 2.89 | 0.23 | 8.9 | 7.00 | 8.22 |
| Clay | % | 23.77 | 6.33 | 44.36 | 7.60 | 26.64 |
| Sand | % | 18.79 | 9.45 | 75.67 | 5.70 | 50.72 |
| Silt | % | 32.79 | 8.32 | 64.16 | 5.21 | 22.62 |
| CCE | % | 42.99 | 6.92 | 34.00 | 00.00 | 17.09 |
| OC | % | 43.93 | 0.25 | 2.60 | 0.06 | 0.57 |
| ESP | % | 28.66 | 1.52 | 12.00 | 0.34 | 2.97 |
| EC | dSm$^{-1}$ | 42.15 | 0.187 | 1.51 | 0.02 | 0.44 |
| CEC | cmolckg$^{-1}$ | 33.25 | 4.76 | 37.82 | 4.97 | 14.33 |
| Gravel | % | 37.72 | 8.72 | 64.00 | 1.00 | 24.72 |

The coefficient of variation (C.V.) is a statistical index of the dispersion of data points around the average. The standard deviation (S.D.) that must always be considered in the context of the mean of the data.

### 3.2. Soil Development

Geomorphological conditions of the study area have the greatest effect on many soil characteristics such as soil depth, soil texture, organic matter, drainage, gravel content, and accumulated calcium carbonate in the soil horizons; and different geomorphic surfaces have affected the development and evolution of soils. The diagnostic surface (ochric) and subsurface horizons (cambic and calcic) are the distinguishing features of the studied soils. The most important soil forming processes are the movement and leaching of calcium carbonate in the soil profile; and the accumulation of organic matter in the surface horizons. The soil profiles were allocated into two orders (i.e., Inceptisols and Entisols), two suborders (i.e., Xerepts and Orthents), three great groups (Haploxerepts, Calcixerepts, and Xerorthents), four subgroups (i.e., Typic Haploxerepts, Typic Calcixerepts, Typic Xerorthents, and Lithic Xerorthents), and seven families (fine loamy, mixed, active, mesic, Litic Xerorthent; loamy skeletal, mixed, superactive, mesic, Typic Calcixerept; loamy skeletal, mixed, superactive, mesic, Typic Haploxerept; sandy skeletal, mixed, mesic, active, Lithic Xerorthent; fine loamy, mixed, superactive, mesic, Typic Calcixerept; loamy skeletal, mixed, superactive, mesic, Typic Xerorthent; fine loamy, mixed, superactive, mesic, Typic Xerorthent). The soils formed in mountain and hill landscapes are Entisols; the reason for this is the high slopes and the lack of stability of the geomorphic surfaces. In contrast, the soils formed in the piedmont landscape are Inceptisols with a dominant calcification process (Table 1).

### 3.3. Selected Auxiliary Variables

Figure 4 shows the relative influence of each auxiliary variable used in the modeling and prediction of land suitability classes. The relative influence of each auxiliary variable within the modelling was calculated in the RF model according to the mean decrease in impurity (or gini importance) mechanism, while in the SVM model the weights reflect the importance of the auxiliary data. In terms of predicting land suitability classes of rain-fed wheat using RF and SVM models, the MrRTF (18.08% and 16.40%, respectively), MrVBF (17.07% and 18.76%, respectively), slope (16.36% and 14.67%, respectively), LS factor (10.81% and 9.97%, respectively), TWI (10.01% and 11.69%, respectively), catchment slope (9.85% and 8.17%, respectively), elevation (8.53% and 10.22%, respectively), and geomorphology map (9.25% and 10.09%, respectively) were most important. For predicting land suitability classes of barley using RF and SVM, the MrVBF (22.88% and 19.43%, respectively), MrRTF (19.59 % and 21.89%, respectively), slope (21.20 % and 20.06%, respectively), LS factor (12.44% and 13.59%, respectively), elevation (9.61% and 11.91%, respectively), and geomorphology map (14.25 % and 13.10%, respectively) were most important. TWI, MrRTF, and MrVBF represent the flat areas where high values indicate low and flat areas. The values of TWI, MrRTF, and MrVBF are higher in geomorphic surfaces such as the alluvial plain and pediment with low slopes (with land suitability class of S3) compared to the geomorphic surfaces of hill and mountain (with land suitability class of N2). In contrast, slope, LS factor, and catchment slope are higher in the hill and mountain geomorphic surfaces that have high slopes compared to other geomorphic surfaces (Figure 3). The larger relative influence of the auxiliary data extracted from the DEM for prediction of land suitability classes demonstrates the impact of topography, as one of the main soil forming factors, on the land suitability of the study region for rain-fed wheat and barley. Topography has significant effects on many of the processes that are important in soil formation. Many studies have found that topography has a considerable impact on soil properties. Nabiollahi et al. [30] showed that the mean soil loss rates were significantly different and higher than other slope classes for the >10% slope class. Conversely, they also showed that the mean soil quality, the >10% slope class, was lowest compared to other slope classes.

Similar results have been reported by other researchers. Nabiollahi et al. [32] used a digital soil mapping technique to assess the spatial variability of soil organic carbon stocks under land use change scenarios in Marivan county of Kurdistan province in western Iran and showed that the most important auxiliary variables were TWI, MrRTF, MrVBF, NDVI index, Band 3 and Band 4 of Landsat 8 ETM. Pouladi et al. [67] mapped soil organic matter content in Denmark using RF and cubist models

and found that the most important auxiliary variables were Blue Spot, TWI, valley depth, MrVBF, Mid-slope position, SL factor, slope gradient, apparent electrical conductivity, elevation, aspect, a ratio vegetation index, and a difference vegetation index. Dang et al. [38] mapped crop suitability areas using a hybrid neural-fuzzy model in the Sapa district of northern Vietnam, and found that slope, elevation, length of flow, ratio of evapotranspiration to precipitation, soil erosion, sediment retention, and water yield were the most important environmental variables.

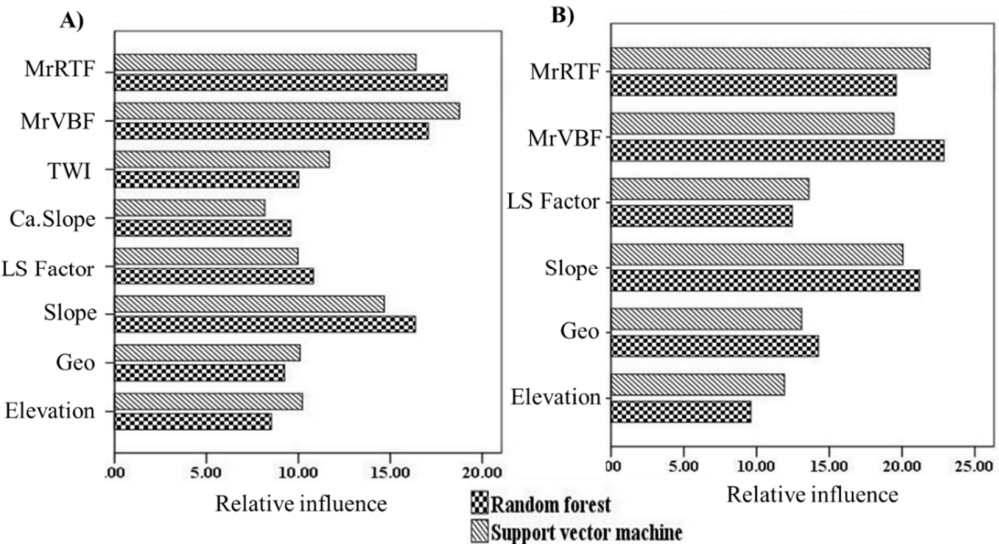

**Figure 4.** The relative influence of each auxiliary variable used in the models for prediction of (**A**) rain-fed wheat land suitability class and (**B**) barley land suitability class. (MrRTF: multi-resolution ridge top flatness; MrVBF: multi-resolution valley bottom flatness index; TWI: topographic wetness index; Ca.Slope: catchment slope; Geo: geomorphology map).

*3.4. Comparison of Different ML Models*

The ability of two ML models (SVM and RF) to predict five land suitability classes of rain-fed wheat and barley in Ghorveh region was tested based on 10-fold cross-validation with 100 replications. Results for the kappa index and overall accuracy for two ML models are summarized in Table 3. The kappa index and overall accuracy for predicting land suitability classes for rain-fed wheat using the RF method (0.77 and 0.79, respectively) were high compared the values for the SVM method (0.57 and 0.63, respectively). Comparable to the land suitability classes of rain-fed wheat, the kappa index and overall accuracy for predicting land suitability classes of barley were higher for RF (0.69 and 0.73, respectively) than for SVM (0.58 and 0.66, respectively). Based on classes of kappa index values defined by [68,69], the RF, and SVM ML models for predicting land suitability class of rain-fed wheat and barley have a strong and moderate ability to predict land suitability classes, respectively. From a statistical point of view, the RF model performed well in terms of prediction ability. Its advantages include the ability to model nonlinear relationships using both categorical and continuous predictor variables [62,70–72] and low bias and variance [73,74]. The RF model is considered a powerful modeling technique for predicting land suitability classes because (i) it is quite robust to noise in predictors, (ii) it shows no overfitting, (iii) it produces predictions with low bias and low variance, (iv) it is quite robust to noise in predictors, and (v) it identifies the most important auxiliary data (55). Therefore, the RF model can be recommended as the best model for the prediction of land suitability class for rain-fed wheat and barley in western Iran. This result is comparable to the findings of other researchers who demonstrated the reliable performance of RF [75–79].

Pahlavan-Rad and Akbarimoghaddam [79] used RF to predict soil texture fractions and pH in Zahak county of Sistan and Baluchestan province in eastern Iran and they reported that the RMSE (root mean square error) values for the sand, silt, clay, and pH maps in validation data were 21.40%, 17.45%,

6.06%, and 0.45, respectively. Camera et al. [80] applied RF and logistic regression (LR) to predict soil classes (world reference soil group) based on world reference base for soil resources (WRB), soil depth, and soil texture classes. They showed that RF performed better than LR. Nabiollahi et al. [31] also observed high performance of RF for modeling soil quality indices using in Kurdistan province in western Iran. Roell et al. [25] applied RF to predict winter wheat yield. They produced a new map of winter wheat yield and compared the results with potential historical yield. Auxiliary data were soil, climate, and topography. Their findings showed that the RF model performed better than the model based only on soil data.

**Table 3.** Error criteria for prediction of land suitability class (SVM: support vector machine; RF: random forest) based on 10-fold cross validation.

| | Kappa Index | | Overall Accuracy | |
|---|---|---|---|---|
| **ML Model** | **Rain-Fed Wheat** | **Barley** | **Rain-Fed Wheat** | **Barley** |
| RF | 0.77 | 0.69 | 0.79 | 0.73 |
| SVM | 0.57 | 0.58 | 0.63 | 0.66 |

*3.5. Land Suitability Class*

The land suitability classes for the studied crops are explained in detail in the following sections:

3.5.1. Marginally Suitable Class (S3)

The areas of this class identified by ML-based and traditional approaches, ranged between 16.50% and 23.07%, for rain-fed wheat and between 18.50% and 26.78% of the studied area for barley (Table 4). The majority of the S3 class has geomorphic surfaces such as river plain (Pi311) and pediment with low slope (Pi221), which are located in the central and south part of study area. These areas had suitable properties such as low slope and elevation, with little or no soil erosion risk, sufficient rates of annual and suitable seasonal and monthly rainfall totals (with the exception of rainfall in the flowering stage). Sunshine hours and optimum temperatures in the phonological stages of rain-fed wheat and barley were also suitable. Land use of the S3 class is cropland and the major limitations are high gravel content, high pH (basic pH), and rainfall in the flowering stages. In these areas most improvement operations have already been applied, therefore, if production is to increase, land improvement operations need to be made such as pH reduction, collecting surface gravel, increasing soil organic matter by farmyard manure, green manure and cover crops cultivation, and irrigating during the flowering stages. Based on the calculated potential yield of rain-fed wheat and barley, yield contents of rain-fed wheat and barley in S3 class can ranged between 2.4–3.6 ton ha$^{-1}$ and 2.2–3.3 ton ha$^{-1}$, respectively. Although, actual yield contents of rain-fed wheat and barley in the study area is lower than these contents because of limitations. Pilevar et al. [81] used integrated fuzzy, AHP, and GIS techniques for land suitability assessment for rain-fed wheat and maize production in 5474.27 ha of saline and calcareous soils located in semi-arid regions, east of Iran. Their results indicated 14.68% (803.75 ha), 78.23% (4282.53 ha), and 7.08% (387.99 ha) of the studied area were highly (S1), moderately (S2), and marginally (S3) suitable for rain-fed wheat production, respectively. Moreover, 2.75% (150.52 ha), 61.51% (3366.99 ha), and 35.74% (1956.76 ha) of the region were highly, moderately, and marginally suitable for maize production, respectively.

3.5.2. Currently Non-Suitable Class (N1)

Based on the ML-based and traditional mapping approaches, the area of this class ranged between 40.32% and 53.50% for rain-fed wheat, and between 37.61% and 59% of the studied area for barley (Table 4). This class is composed of geomorphic surfaces such as fan with moderate slope (Pi111), and pediment with moderate and high slope (Pi211, and Pi212). Current land use of the N1 class is cropland and the major limitations of the class are moderate and high slopes and rainfall in the

flowering stage. In these areas most potential improvement operations have already been applied, therefore, to increase production and prevent land degradation, land improvement operations such as leveling terracing and increasing soil organic matter by farmyard manure, green manure, and cover crops cultivation, are needed. Based on this calculated potential yield of rain-fed wheat and barley, yield contents of rain-fed wheat and barley in currently non-suitable class can range between 1.8–2.4 ton ha$^{-1}$ and 1.1–2.2 ton ha$^{-1}$, respectively. Although, actual yield contents of rain-fed wheat and barley in the study area is lower than these contents because of limitations.

In an investigation of land suitability of Golestan Province, located in the northeast of Iran, for rain-fed farming, Kazemi and Akinci [82] showed that salinity, organic matter, soil erosion, soil texture classes, and the autumn, spring, May and June rainfall, and slope were limiting factors for rain-fed farming. They also suggested that land improvement practices such as increasing soil organic matter, salinity reduction, conservation tillage, and supplementary irrigation are used.

### 3.5.3. Permanent Non-Suitable Class (N2)

Based on ML-based and traditional approaches, this class covered 30% and 36.61% of the studied area for rain-fed wheat and barley, respectively, (Table 4). Land use of the N2 class is rangeland and has severe limitations such as shallow depth, rock outcrops, and very high slopes for cropland. This class is comprised of geomorphic surfaces of hills (Hi111, Hi121, Hi211, Hi221, Hi231, Hi232, Hi241, and Hi251) and mountains with high and very high slopes (Mo111, Mo121, and Mo131), and it is located in the north, northeast, and northwest of the study area.

Ostovari et al. [83] prepared a land suitability map of 51,831 ha in the northwest part of East Azerbaijan province, Iran for rapeseed farming using GIS and multi-criteria decision-making analysis. Their results showed that 0.81% (420.8 ha), 42.33% (21,940.2 ha), and 11.78% (6104 ha) of the studied area fell into the highly suitable (S1), moderately suitable (S2), and marginally suitable (S3) classes, respectively, and 39.72% (20,586.4) and 0.95% (492.1 ha) of the studied area was currently not-suitable and permanently not-suitable for rapeseed production. Soil parameters and topography factors (elevation and slope) were the most important in their study area.

**Table 4.** Land suitability classes of the study area.

| Land Suitability Class | ML-Based Approach | | | | Traditional Approach | | | |
|---|---|---|---|---|---|---|---|---|
| | Rain-Fed Wheat | | Barley | | Rain-Fed Wheat | | Barley | |
| | Area (ha) | Area (%) | Area (ha) | Area (%) | Area (ha) | Area (%) | Area (ha) | Area (%) |
| Marginally suitable (S3) | 10,725 | 16.50 | 12,025 | 18.50 | 14,995.50 | 23.07 | 17,407 | 26.78 |
| Currently non-suitable (N1) | 34,775 | 53.50 | 37,050 | 57.00 | 26,208 | 40.32 | 24,446.50 | 37.61 |
| Permanently non-suitable (N2) | 19,500 | 30.00 | 15,925 | 24.50 | 23,796.50 | 36.61 | 23,146.50 | 35.61 |
| Sum | 65,000 | 100 | 65,000 | 100 | 65,000 | 100 | 65,000 | 100 |

### 3.6. Comparing Traditional and ML-Based Approaches

The kappa index and overall accuracy for comparing the accuracy between traditional and ML-based land suitability maps based on external validation are shown in Table 5. The ML-based land suitability maps for rain-fed wheat and barley had higher kappa index values and overall accuracy (0.77 and 0.79; 0.69 and 0.73, respectively) compared to traditional land suitability maps (0.45 and 0.50; 0.43 and 0.47, respectively). Based on the class limits of the kappa index established by [68,69], ML-based land suitability maps of rain-fed wheat and barley have strong levels of accuracy and traditional land suitability maps of rain-fed wheat and barley have moderate levels of accuracy. ML-based and traditional approaches are similar in that they both make use of relationships between soil properties and more readily observable land surface properties. In the traditional approach using geomorphologic units as land suitability delineations may lead to unsatisfactory results in estimation of quantity and type of existing limitations. Traditional land suitability maps are limited by the scale of the base map, their inability to represent continuous land suitability classes, and spatial variation. Moreover,

the traditional approach is time consuming and costly [20]. The ML-based approach map is less influenced by these constraints, and is preferred for handling the usual land suitability assessment design. Zeraatpisheh et al. [55] also compared digital and conventional soil mapping approaches to predict soil classes in the semi-arid region of Borujen, central Iran. They showed that the costs of both soil mapping approaches were almost equal, although digital soil mapping approaches had higher map purity, kappa index values, and diversity than the traditional approach.

**Table 5.** Error criteria for comparing ML-based and traditional land suitability maps for the external validation based on random forest model.

|  | Kappa Index | | Overall Accuracy | |
| --- | --- | --- | --- | --- |
| **Approach** | **Rain-Fed Wheat** | **Barley** | **Rain-Fed Wheat** | **Barley** |
| ML-based land suitability map | 0.77 | 0.69 | 0.79 | 0.73 |
| Traditional land suitability map | 0.45 | 0.43 | 0.50 | 0.47 |

ML-based maps of land suitability class for rain-fed wheat and barley using the RF model were prepared and are shown in Figure 5. Approximately 30.00%, 53.50%, and 16.50% of the study area were designated as classes N2, N1, and S3 for rain-fed wheat, respectively (Table 4 and Figure 5). Approximately 24.50%, 59.00%, and 18.50% of the study area were designated as N2, N1, S1 and S2 classes for barley, respectively (Table 4 and Figure 5).

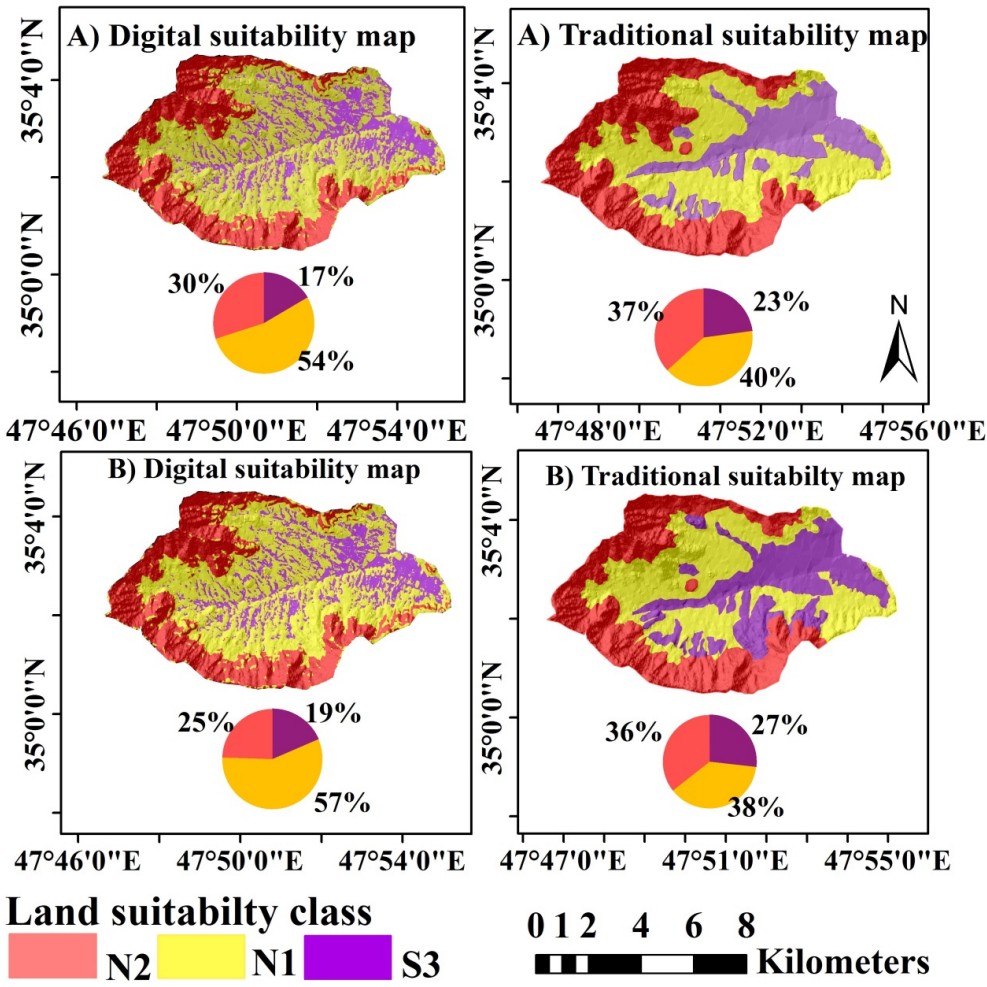

**Figure 5.** ML-based and traditional maps of (**A**) rain-fed wheat land suitability class and (**B**) barley land suitability class based on random forest model.

Traditional maps of land suitability class of rain-fed wheat and barley are also shown in Figure 5. Like the ML approach, land suitability classes for rain-fed wheat the N2, N1, and S3 classes accounted for 36.61%, 40.32%, and 23.07%, of the study area, respectively (Table 4) and for barley, 35.61%, 37.61%, and 26.78%, of the study area was identified as classes N2, N1, and S3, respectively (Table 4). These results from the traditional approach show an overestimation of the N2 and S3 classes compared to the ML-based approach. In contrast, the area of N1 was low in the traditional approach compared to the ML-based approach. Areas of N2 and S3 were 6.61%–11.11% and 6.07%–8.87%, high based on the traditional approach compared to the ML-based approach for rain-fed wheat and barley, respectively. Conversely, the area of N1 was approximately 13.18%–21.39%, low based on the traditional approach compared to the ML-based approach for rain-fed wheat and barley, respectively.

## 4. Conclusions

This study assessed land suitability for rain-fed wheat and barley on agricultural land in Kurdistan province, Iran. This was done using the FAO land suitability assessment framework, parametric method, and two mapping approaches namely ML-based and traditional methods. The MrRTF, MrVBF, slope, LS factor, TWI, catchment slope, elevation, and geomorphology map were the most important auxiliary data for predicting land suitability class of rain-fed wheat and barley. Based on kappa index results, RF was the better ML model for predicting land suitability class of rain-fed wheat and barley compared to the SVM approach. RF has several advantages over other statistical modeling approaches and is considered a powerful modeling technique for predicting land suitability classes. Comparison between traditional and ML-based approaches also showed that the ML-based approach identified larger areas of N2 and S3 classes and smaller areas of the N1 class than the traditional approach.

The traditional approach is time consuming and costly. In contrast, the ML-based approach map is less influenced by these constraints and is preferred for handling the usual land suitability assessment design. This is particularly true in a data-poor region such as Iran, where little soil information is available. Therefore, machine learning and auxiliary data can be an attractive approach for land suitability assessment at the large scale.

In general, the study area results showed low suitability or that it was not suitable for cropland due to of limitations of rainfall in the flowering stage, severe slopes, shallow soil depth, high pH, and gravel. These limitations are the main reasons why the actual yield of rain-fed wheat and barley in the study area is lower than potential yield of those crops. Hence, to improve the suitability of the study area for croplands and increase its production, land improvement operations such as terracing, decreasing pH, increasing soil organic matter by farmyard manure, green manure and cover crops cultivation, supplementary irrigation, and gathering gravel are needed. This study provided useful information that can be applied to quantify the implication of management policies in Kurdistan province and other similar regions.

**Author Contributions:** Conceptualization—K.N.; R.T.-M.; Methodology—K.N.; R.T.-M.; L.R.; Software—R.T.-M., K.N.; L.R.; Analysis—R.T.-M., K.N., R.K., R.K., T.S.; Investigation— R.T.-M., K.N., L.R.; Data curation—K.N., L.R.; Writing—Original Draft Preparation— R.T.-M., K.N., R.K., T.S.; Visualization— R.T.-M., K.N., L.R. All authors have read and agreed to the published version of the manuscript.

**Funding:** The University of Kurdistan funded this research and Ruhollah Taghizadeh-Mehrjardi was funded by the Alexander von Humboldt Foundation grant number Ref 3.4-1164573-IRN-GFHERMES-P.

**Acknowledgments:** This study was part of a master project entitled "Assessing the effect of soil quality and type on rain fed wheat yield using soil quality index". Thomas Scholten thanks the German Research Foundation (DFG) for supporting this research through the Collaborative Research Center (SFB 1070) 'ResourceCultures' (subprojects Z, S and B02). He is also supported by the DFG Cluster of Excellence "Machine Learning—New Perspectives for Science", EXC 2064/1, project number 390727645.

**Conflicts of Interest:** The authors declare no conflict of interest.

# Appendix A

**Table A1.** Geomorphology map key and the major soil sub-groups, land index and current and potential future land suitability class per geomorphological surface.

| Code | Landscape | Landform | Lithology | Geomorphological Surface. |
|---|---|---|---|---|
| Hi111 | Hill land | Eroded high hill | Monzodiorite, quartz monzonite | Continuous hill with high topography |
| Hi121 | | | Foliated and brecciated | Continuous hill with high topography |
| Hi211 | | Eroded moderate hill | Monzodiorite, quartz monzonite | Continuous hill with high topography |
| Hi221 | | | Granite, amphibole, granodiorite | Continuous hill with high topography |
| Hi231 | | | Foliated granite | Continuous hill with high topography |
| Hi241 | | | Cordierite schist(spotted schist) | Continuous hill with high topography |
| H251 | | | Limestone | Continuous hill with high topography |
| Pi111 | Piedmont | Alluvial fan | Alluvial fan | Active fan |
| Pi211 | | Pediment | High level terraces | High slop |
| Pi212 | | | | Moderate slop |
| Pi221 | | | Dikes of aplite granite | Low slop |
| Pi311 | | River plain | Lowest alluvial plain | Cultivated flat |
| Mo111 | Mountain | Rock outcrop | Monzodiorite, quartz monzonite | Eroded rock surface |
| Mo121 | | | diorite and gabbro | Eroded rock surface |
| Mo131 | | | Foliated and brecciated | Eroded rock surface |

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
