# Peer review of "Land Suitability Assessment and Agricultural Production Sustainability Using Machine Learning Models"

_agronomy, doi:10.3390/agronomy10040573_

Round 1

Reviewer 1 Report

Comments to the Author

Reviewer’s Comments on the Paper Entitled “Land suitability assessment and agricultural production sustainability using machine learning models” (Manuscript ID. Agronomy-771472)

I have gone through the manuscript of the above paper. This paper focuses on land suitability assessment for wheat and barley by using the FAO "land suitability assessment framework" and parametric method for 65 km2 of agricultural land in a semi-arid region of Kurdistan Province, Iran. The authors used data from 100 soil profiles including genetic layers, the physical-chemical properties of the soil, topography, and climate. Support vector machines and random forests were implemented to determine relations between auxiliary variables and land suitability classes. The authors concluded that the major limitations of the study area to cultivating rain-fed wheat and barley were rainfall at the flowering stage, severe slopes, shallow soil depth, high pH and large gravel content. Similarly, they stated that to increase production and create a sustainable agricultural system it is suggested that land improvement operations such as terracing, decreasing pH, supplementary irrigation and gathering gravel be employed.

The study is interesting. The authors have synthesized the information nicely. The scope of the paper is well presented. Similarly, the question of the study is clear. The language used for the study is good but needs improvements. Here I provided some minor comments that need to be considered during the revision.

Minor comments:

  1. Line 29. Add a “to” before “determine…”.
  2. Line 33. Add a “the” before “traditional…”.
  3. Line 34. Add a “the” before “class of N1…”.
  4. Line 35. Replace “was” with “were”.
  5. Line 57. Add an “s” to “… evaluation” and a “the” before qualitative.
  6. Please check similar language mistakes for the entire paper.
  7. Why do you think support vector machines and random forests need to be used and why they are beneficial? Why not something else?
  8. Line 90 and 91. What does this objective bring to science? Why do we need a map of that specific area? I would suggest improving the objective in a way to state what new things this work can bring to science?
  9. Line 92. To have such an objective, I think the introduction should give background for this objective with a detailed explanation of why this is important to science.
  10. Line 107 to 123. I would convert this section to a paragraph.
  11. The author has very nice graphs, figures, and a deep understanding of statistical approaches.
  12. Please work on the language a little bit. It is good but it can be better with some improvements.
  13. Please strengthen the discussion and especially the conclusion. I believe there is much more to conclude from his work.
  14. Nice work!

Author Response

We are grateful for the valuable comments of the editors and reviewers and have revised the manuscript accordingly. All changes are marked in the marked_agronomy document. Here we reply to the comments of the editors and reviewers in detail.

Reviewer 1

I have gone through the manuscript of the above paper. This paper focuses on land suitability assessment for wheat and barley by using the FAO "land suitability assessment framework" and parametric method for 65 km2 of agricultural land in a semi-arid region of Kurdistan Province, Iran. The authors used data from 100 soil profiles including genetic layers, the physical-chemical properties of the soil, topography, and climate. Support vector machines and random forests were implemented to determine relations between auxiliary variables and land suitability classes. The authors concluded that the major limitations of the study area to cultivating rain-fed wheat and barley were rainfall at the flowering stage, severe slopes, shallow soil depth, high pH and large gravel content. Similarly, they stated that to increase production and create a sustainable agricultural system it is suggested that land improvement operations such as terracing, decreasing pH, supplementary irrigation and gathering gravel be employed.

The study is interesting. The authors have synthesized the information nicely. The scope of the paper is well presented. Similarly, the question of the study is clear. The language used for the study is good but needs improvements. Here I provided some minor comments that need to be considered during the revision.

Thank you for your very valuable and positive feedback.

Line 29. Add a “to” before “determine…”:

This has been done.

Line 33. Add a “the” before “traditional…”

This has been done.

Line 34. Add a “the” before “class of N1…”.

This has been done.

Line 35. Replace “was” with “were”.

This has been done.

Line 57. Add an “s” to “… evaluation” and a “the” before qualitative.

This has been done.

Please check similar language mistakes for the entire paper.

Thank you for raising this point. This has been done for the entire paper.

Why do you think support vector machines and random forests need to be used and why they are beneficial? Why not something else?

We thank the reviewer for pointing out this issue. We selected two machine learning models including random forest (RF) and support vector machine (SVM) due to their successful applications in earlier studies [29-35] and their relatively good accuracy, robustness and ease of use. Importantly, it has been proved that both RF and SVR models work well when there is no massive availability of data. Additionally, it is important in this work to explain the complex relationships between the predicted land suitability class and the auxiliary variables, which can be inferred by the exploring the importance of auxiliary variables.

Line 90 and 91. What does this objective bring to science? Why do we need a map of that specific area? I would suggest improving the objective in a way to state what new things this work can bring to science?

Thank you for raising this point. The objective part of our paper is modified:

Although Kurdistan province is one of the main agriculturally productive regions of Iran and holds an important role in the country’s crop production rank, the mean yield of rain-fed wheat and barley in these regions are lower than 800 kg ha-1 [40]. Land suitability maps can classify the areas that are highly suitable for the cultivation of the two main crops and can help to increase their production. But such information is commonly scarce in these semi-arid regions. Therefore, the main objective of this study is to assess the land suitability for two main crops based on the FAO "land suitability assessment framework". Furthermore, in this study, we focus on using machine learning models to predict the spatial distribution of land suitability classes in the most economically feasible way and explore if it works better than the traditional approach¾the most common approach to produce land suitability class maps in Iran.

Line 92. To have such an objective, I think the introduction should give background for this objective with a detailed explanation of why this is important to science.Thank you for the comment. We extended the introduction to give background the objectives of our work. Importantly, we modified the machine learning definition parts in the introduction.

Line 107 to 123. I would convert this section to a paragraph.

Thank you for the comment. However, we would like to keep the procedures as it is. But we modified the writing to make it easier to read.

Please work on the language a little bit. It is good but it can be better with some improvements.

Thank you for raising this point. This has been done for the entire paper.

Please strengthen the discussion and especially the conclusion. I believe there is much more to conclude from his work.

Thank you for the comment. We extended the discussion and conclusion parts. This has been done (414 to 419, 428 to 431, 456 to 462, 205 to 513, 544 to 555).

Reviewer 2 Report

see the attached pdf document (comments.pdf).

Author Response

Reviewer 2

In this paper, the authors propose a study on land suitability assessment for two main crops for agricultural land in a semi-arid region of Kurdistan Province, Iran. Land suitability assessment is essential for increasing production and planning a sustainable agricultural system but such information is commonly scarce in the semi-arid regions of Iran.

Two ML-based methods are used to generate land suitability class maps for the main crops and the classification results of two methods are compared to the traditional land suitability maps. The use of ML-based techniques helps to improve the interpretability of the study at hand in this research.

I encourage the publication of this work because of the interesting specific area of application of the mentioned ML-based methods; nevertheless, I strongly recommend a revision of this article providing additional suitable explanations for some technical details and performance measures as I suggest below with details. In this perspective, a revision of the article will produce a good reference for the user to whom this work is dedicated.

Thank you for your very valuable and positive feedback. We revised the paper according to the comments. Importantly, we extended the method section and technical parts.

Abstract: well done and structured in my opinion; I only suggest to summarize more, i.e., using more or less 200 words. Maybe lines 30-34 could be omitted because they include numeric details of results that can be summarized in an abstract or rewritten in brief

Thank you for the comment. The abstract is summarized in 225 words.

Figures 1, 3, 5: assuming that it is not possible to improve the graphic resolution I suggest to enlarge them when possible just to appreciate better the axes

Thank you for the comment. The resolution of figures are improved.

Figure 2, line 128: check the acronym: do you mean SVR or SVM? Assuming that you are solving a classification problem I think it's SVM as you also specify in the rest of the paper. SVR could be the acronym for Support Vector Regression, so revise that. Apart from that, Figure 2 is very useful for comprehension in general

Great point. This has been corrected to SVM.

Line 180: SVR appears another time (see the previous comment). One very important thing that in my opinion is missing here and that I recommend to add is a brief explanation about the choice of ML-based methods you use. I mean, why RF and SVM have been taken into account among other algorithms? Maybe this choice is due to the fact that there's no massive availability of data and/or the fact that the interpretability of results in terms of variable selection and the quantification of their importance for prediction is mandatory in this work. I suggest to make clear the reason why you use the techniques mentioned for your specific task; this will be very helpful for an interested reader who has a similar problem;

We thank the reviewer for pointing out this issue. We selected two machine learning models including random forest (RF) and support vector machine (SVM) due to their successful applications in earlier studies [29-35] and their relatively good accuracy, robustness and ease of use. Importantly, it has been proved that both RF and SVM models work well when there is no massive availability of data. Additionally, it is important in this work to explain the complex relationships between the predicted land suitability class and the auxiliary variables, which can be inferred by the exploring the importance of auxiliary variables [Lines 92-98].

Table 1: I suggest to center it on the page just to avoid the empty space at the bottom;

We thank the reviewer for pointing out this issue. We moved the Table to the Appendix. This table gives the definition of geomorphic surfaces.

page numbering issue: I don't know why but from line 202 the page counter seems to restart, so from line 202 the page is page 1 instead of 7, the following page is page 2 instead of 8 etc. I suggest to check it and solve (if it is done in latex, check if the problem is due to some packages used for tables or some instructions at that line);

We thank the reviewer for pointing out this issue. After moving the Table to the Appendix, the page number is corrected.

line 219: the use of Boruta algorithm in R is very interesting in my opinion to solve this problem because it provides an easy and immediate interpretation of results in terms of variable selection. I suggest spending few words in this part to describe it better, especially explaining how the "importance" is computed for each predictor (you mention it in line 226 but without explanations, so try to include a brief description from the CRAN package documentation for example or citing something). In this way the results you visualize in the rest of the paper will be much more clear (e.g., the relative influence in Figure 4 etc.)

We thank the reviewer for pointing out this issue. We extended the definition of the algorithms [Lines 223-225]. We also added some sentences to the auxiliary influence section [Lines 332-335]. 

line 236: just put the bullet point on a new line

Thanks. We modified the section numbers in the method. 

line 242: there is additional space in "forest package__of";

This has been deleted.

line 246: here, in my opinion, the most important point that I suggest to modify: please include an explanation for kappa index and overall accuracy because they are used in the rest of the paper and their comprehension is crucial to get into the results

We thank the reviewer for pointing out this issue. We extended the definition of the kappa and overall accuracy [Lines 242-261].

11.line 273: additional space "biomass__on overcast

This has been deleted.

  1. line 274: additional space "biomass__on clear days";

This has been deleted.

  1. line 276: Bn instead of bn;

This has been done.

  1. line 287: I suggest to spend few words to explain the harvest index

This has been added. “harvest index (ratio of grain yield to above ground biomass)”

  1. line 299: even though it is clear that CV stands for Coefficient of Variation, I suggest to specify it also adding an explanation (formula or a sentence). Moreover, specify that S.D. stands for Standard Deviation in Table 2;

Great point. This has been added. Please see the Table 2.

line 324: additional space: dominant__calcification process

This has been deleted.

Figure 4: considering what said before, I recommend to spend few words to explain how the relative influence reported on x-axis is computed. It is essential to obtain a clear interpretation of your results;

This has been added. We added some sentences to the auxiliary influence section [Lines 332-335]. 

I suggest to explain what is the RMSE with a formula or in brief. It is a well-known index for evaluations of interest but it is always good to specify it;

This has been done. “(Root mean square error)”

what id MLR? Does it stand for Multiple Linear Regression? Please specify it

This has been corrected. It was typo and SVM is correct.

I only suggest to highlight the row of RF in order to stress that it is the main result. I mean, you can write it in bold or if you use latex, something like: \cellcolor{gray}{your text}. It can be done also for other tables in the paper, for example table 5. It can help the reading but it is only a suggestion;

This has been done.

line 466: additional space: barley,__

This has been deleted.

In general, in my opinion, this work is well written as well as the associated research. In particular, I consider remarkable the use of some specific packages for the purpose at hand, such as Boruta and so on. In general, I recommend adding some additional explanations for technical details related to the use of these packages and metrics of evaluation in order to improve the comprehension and to enhance the entire work. Moreover, I recommend adding few lines to explain the reasons why some ML-based methods are used instead of others for the problem at hand. In my opinion, it is necessary to revise these things before publication. As for the rest, as usual, I suggest performing only quick minor revisions such as typos etc. as reported above.

Thank you for your very valuable and positive feedback. We revised the paper according to the comments. Importantly, we extended the method section and technical parts.